# Analysis of Pollution Characteristics and Sources in Surface Water in Typical Crop-Producing Areas of Qinghai Province

**DOI:** 10.3390/ijerph192416392

**Published:** 2022-12-07

**Authors:** Pengtao Chen, Furong Fu, Jie Li, Jingui Wang, Yang Sun, Ruigang Wang, Lixia Zhao, Xiaojing Li

**Affiliations:** 1Agro-Environmental Protection Institute, Ministry of Agriculture and Rural Affairs, Tianjin 300191, China; 2College of Agriculture and Animal Husbandry, Qinghai University, Xining 810016, China

**Keywords:** agricultural areas, Qinghai Province, surface water, pesticides, nitrogen and phosphorus pollution, source apportionment

## Abstract

Currently used pesticides and organochlorine pesticides (OCPs), nitrogen and phosphorus were analyzed in surface water from 26 sampling sites of agricultural areas in Qinghai Province to elucidate their pollution characteristics and sources. The results showed that most of these currently used pesticides, with the exception of chlorpyrifos, were generally not detected. However, two OCPs were commonly detected in surface water from four typical crop-producing areas. The residual concentrations of hexachlorocyclohexanes (HCHs) and dichlorodiphenyltrichloroethanes (DDTs) measured 0~1.68 ng/L and 0.41~2.41 ng/L, respectively, in the water from the four crop-producing areas. The residues of these two OCPs pesticides were much lower than the standard limit of surface water environmental quality. The main forms of HCHs and DDTs were *β*-HCH and *pp’*-DDE, respectively, indicating that the residues of HCHs and DDTs in the surface water of the four crop-producing areas in Qinghai were mainly derived from historical drugs that had degraded for a long time. The average concentrations of TN, NO_3_^−^-N and NH_4_^+^-N in the surface water of 26 sampling sites of four typical crop areas in Qinghai Province were 2.95, 1.71 and 0.17 mg/L, respectively. According to the national surface water environmental quality standards, TN concentrations in 57.7% of these sampling sites exceeded the Class V water standards. The average concentration of NO_3_^−^-N was more than 70 times that of NH_4_^+^-N. Nonetheless, there were no significant differences in the concentrations of TN, NO_3_^−^-N and NH_4_^+^-N in the four crop-producing areas. The concentrations of NO_3_^−^-N and NO_3_^−^-N in the surface water were positively correlated with the TN concentration (*p* < 0.05), indicating that the sources of nitrogen in the surface water were relatively consistent. The average value of TP concentrations in the surface water from these sampling sites was 0.034 mg/L, with no significant differences among different producing areas. The N/P values in surface water from the four crop-producing areas of Qinghai Province had a range of 9.2~302. Phosphorus was the limiting factor for the proliferation of plankton in water. Reducing the input of phosphorus in these areas may be the key to preventing the deterioration of water quality. Significant negative and positive correlations exist between HCHs and nitrate nitrogen, and total phosphorus, respectively, which may be attributed to the proliferation of degrading microorganisms caused by the eutrophication of water. The research results will help to identify the characteristics and sources of surface water pollution in the crop-producing areas of Qinghai Province, and provide data support for Qinghai Province to build an export area for green organic agricultural and livestock products.

## 1. Introduction

Current agricultural production is still largely dependent on the use of pesticides. More than 80% of pesticides are applied to the soil and water environment, and the environmental risks brought by the input of pesticides cannot be ignored [1]. OCPs, especially HCHs and DDTs, which have been used as insecticides for decades, were historically widespread. Before they were banned in 1983, the productions of these two pesticides reached 4.9 million tons and 400,000 tons, respectively, in China [2]. As a result, a large amount of pesticides remaining in soil entered the water body through surface runoff, volatilization and dry and wet deposition [3]. Due to the high biological enrichment of HCHs and DDTs, they can eventually be increased by more than ten million times by water organisms [4]. Therefore, the monitoring and evaluation of OCPs in water bodies is of great significance for the risk assessment of non-point sources in watershed. The pollution of OCPs in rivers, lakes and other environmental media has attracted continuous attentions from researchers at home and abroad, and has always been a research hotspot in the environment field [5,6,7,8,9]. Many problems, such as overexploitation of the environment by human, excessive use of chemical fertilizers in agricultural production and disordered discharge of livestock manure due to the surge of animal husbandry scale, have led to the serious increase of nitrogen and phosphorus load in agricultural basins. Water eutrophication caused by the increase of nitrogen and phosphorus concentrations in water has become an important environmental problem worldwide, posing a major threat to water’s ecological stability and water health [10,11,12].

Qinghai Province is located in the northeastern part of the Qinghai Tibet Plateau, and has a total land area of 722,300 square kilometers, ranking fourth in China. The Qinghai Tibet Plateau is known around the world as the “Chinese water tower”, and Qinghai Province is very sensitive to global climate change and plays an extremely important ecological role. Sanjiangyuan in the southwest is the birthplace of three rivers, i.e., the Yangtze River, the Yellow River and the Lancang River. The cultivated land is mainly distributed in the warm Hehuang Valley in the east, the lower Qaidam Basin in the west and the plateau valley. In recent years, the sown area of crops in Qinghai Province remained stable at 590,000 hectares, mainly concentrated in the prefectures of Haidong, Xining, Hainan and Haixi, accounting for 85% of the total cultivated land area. From 2009 to 2019, the output of grain crops (wheat, coarse grains, cereals and tubers) in Qinghai Province stood at 2 million tons/year. With the rapid development of agriculture in Qinghai, there is also serious incoordination between input and output, such as excessive extensive agricultural cultivation, unreasonable input of fertilizer, medicine and other resources, backward development of agricultural facilities and a low utilization rate of crops, which lead to serious environmental problems [13].

The research on pollution characteristics and source analysis of currently used pesticides, OCPs, and nitrogen and phosphorus of water in Qinghai agricultural area is the basis for effectively evaluating the risks to the water ecosystem and to human health, and for formulating targeted prevention and control strategies in a timely manner. However, up to now, research on pesticides and nitrogen and phosphorus pollution in this region is still focused on residue analysis and pollution assessment on soil and crops [14,15,16,17], thus relevant research data on water pollution assessment remains lacking. Therefore, in order to systematically evaluate the current situation of pesticides, and of nitrogen and phosphorus pollution in the surface water of characteristic crop-producing areas in Qinghai Province, to identify the impact of current agricultural and animal husbandry activities on water quality, and to assess the risk of water pollution on ecosystem and human health, it is urgent to ascertain the pollution characteristics and sources of pesticides, and of nitrogen and phosphorus in regional water. This paper evaluates the water quality and sources of pollution in the region by monitoring the concentrations of 13 currently used pesticides and 10 OCPs in the surface water of four kinds of characteristic crops in Qinghai Province, i.e., rapeseed, garlic, lycium chinensis and hulless barley, as well as the physical and chemical index characteristics of nitrogen and phosphorus, in order to provide data support for Qinghai Province to build a green organic agricultural and livestock product export ground [18,19].

## 2. Materials and Methods

### 2.1. Collection of Water Body Samples

In order to gain a comprehensive understanding of the quality of water bodies in the production areas of special crops in Qinghai Province, we collected seven, four, nine and six surface water samples from the production areas of rape, garlic, lycium chinensis and hulless barley, respectively. The above mentioned, currently used pesticides, typical organochlorine pesticides used historically, as well as pH, COD and soil nitrogen and phosphorus were all tested for. The distribution of the specific sampling points is shown in Figure 1.

Referring to the national standard “Collection and Preservation of Water Samples” (GB/T 5750.2-2006), we selected surface water or irrigation channels adjacent to farmland as sampling points. Surface water was sampled from ditch systems, and water samples at 0.3–0.5 m below the water surface were collected from lakes, reservoirs (ponds) and rivers. When sampling, 500 mL of water was collected from each sampling point and poured into a polyethylene sample bottle. After being tightly capped, the bottles were placed in a cooler with ice packs and brought to the laboratory within 24 h. These water samples were divided into two parts, one was stored in a −20 °C refrigerator for the determination of pesticide residue, and the other part was stored in a refrigerator at 4 °C in order to test pH, COD, nitrogen, phosphorus and other indexes of water.

### 2.2. Analytical Test Methods for Pesticides

#### 2.2.1. Preparation of Test Samples

Based on the partial optimization and modification of the method of Yang Song et al. [20], the preparation method was specifically expressed as follows: The Cleanrt^®^-PEP solid phase extraction column was first rinsed with 4 mL of hexane to remove impurities from the packing material, then activated with 6 mL of methanol and 5 mL of deionised water, in turn. Next, 300 mL of water was passed through the column at a rate of 2 mL/min using a solid phase extractor. The column was then washed with 4 mL of pure water and eluted with 6 mL of acetonitrile. The eluate was collected and 5 mL of the eluate was transferred to a 10 mL centrifuge tube, blown to dryness with nitrogen and then re-dissolved in 1 mL of hexane. The content of organochlorine pesticides, HCHs, DDTs and three pesticides, trifluralin, cypermethrin and cyfluthrin, was determined using gas chromatography with a micro-electron capture detector (GC-μECD). The residues of 10 currently used pesticides, including carbendazim, thiamethoxam, imidacloprid, acetamiprid, thiophanate-methyl, tebuconazole, dimethomorph, difenoconazole, chlorpyrifos and dicamba, were determined by LC-MS/MS, after mixing with 0.5 mL of eluate in a 2 mL centrifuge tube and passing through a 0.22 μm filter membrane.

#### 2.2.2. Instrumental Test Methods

GC-μECD: HCHs, DDTs, trifluralin, cypermethrin and cyfluthrin were analyzed using Agilent 7890A gas chromatography (GC), which was equipped with a micro-electron capture detector (μECD) and a HP-5 MS capillary column (30 m × 0.32 mm × 0.25 μm). High purity nitrogen (99.999%) was used as the carrier gas at a flow rate of 3.0 mL/min. The temperatures for the inlet and detector were set to 250 °C and 300 °C, respectively. The column incubator temperature was initially set to 80 °C, held for two minutes and then increased to 170 °C at a rate of 25 °C/min. The temperature in the oven was programmed as follows: the initial temperature was set at 80 °C, held for 2 min, and then increased to 170 °C at 25 °C/min, then to 250 °C at a rate of 5 °C/min, held for 2 min and then increased to 280 °C at a rate of 25 °C/min and finally held for 2 min. The injection volume was 1 μL in pulse splitless mode. The test chromatogram is shown in Appendix A.

LC-MS/MS: Ten currently used pesticides mentioned above were determined by a Shimadzu Ultra Performance Liquid Chromatography system coupled with a SCIEX 4500QTRAP tandem mass spectrometry (LC-MS/MS) operated in negative or positive electrospray ionization (ESI) mode. The separation of these metabolites was performed on a Phenomenex Kinetex^®^ Biphenyl 100A (100 × 2.1 mm, 2.6 μm). The mobile phase was a mixture of 0.01% aqueous formic acid (phase A) and chromatographically pure methanol (phase B) for gradient elution: 0–0.5 min, 80% A + 20% B; 4.0–6.5 min, 5% A + 95% B; 6.6–9.0 min, 80% A + 20% B. The injected volume of extracts was 10 μL and the oven temperature was 40 °C. The Multiple Reaction Monitoring (MRM) mode was used to determine these pesticides. More details are provided in Appendix A and the test chromatogram is shown in Appendix A.

#### 2.2.3. Quality Control and Assurance

To ensure the accuracy and precision of the experimental data, a recovery study was conducted. The concentrations fortified with pesticides mentioned above were 0.01 μg/L, 0.1 μg/L and 1.0 μg/L, respectively. Five replicates for each concentration were set up. The recovery ranges of the GC-μECD and the LC-MS/MS methods in all spiked levels were 79.55–102.2% and 83.74–97.6%, respectively. The corresponding elative standard deviations were 2.08–9.28% and 3.52–7.34%, respectively. The limits of quantification for these pesticides including the OCPs and the currently used pesticides were all 0.1 ng/L. The above spiked recovery data indicate that the analytical method can meet the requirements for the analysis of pesticide residues in farmland water.

### 2.3. Test Methods for Physical and Chemical Indicators of Water Bodies

Determination of pH in water bodies followed the glass electrode method [21], chemical oxygen consumption (COD) was determined using the rapid potassium dichromate method [22], total nitrogen (TN) was determined using the alkaline potassium persulphate digestion UV spectrophotometric method [23], ammonium nitrogen (NH_4_^+^-N) was determined using the nano-reagent spectrophotometric method [24], and nitrate nitrogen (NO_3_^−^-N) and total phosphorus (TP) were determined via the UV spectrophotometric method [25] and the ammonium molybdate spectrophotometric method [26], respectively.

### 2.4. Data Processing and Analysis

The data processing and correlation analysis of this experiment were carried out using SPSS 17.0 statistical software, combined with Origin 8.5 and SPSS17.0 software for graphical plotting and analysis of pesticide residue concentrations and water body physicochemical index data.

## 3. Results and Discussion

### 3.1. Characteristics of Pesticide Pollution in Surface Water and Analysis of Organochlorine Pesticide Sources in Typical Crop-Producing Areas of Qinghai Province

#### 3.1.1. Pollution Characteristics of Currently Used Pesticides

Based on the field survey and literature reviews, we selected six insecticides (chlorpyrifos, acetamiprid, thiamethoxam, imidacloprid, cyfluthrin and cypermethrin), five fungicides (thiophanate-methyl, carbendazim, tebuconazole, dimethomorph and difenoconazole) and two herbicides (dicamba and trifluralin) as objective pollutants, to evaluate the pollution characteristics of currently used pesticide in surface water in typical crop production areas of Qinghai Province. The residual results are listed in Appendix A. The results show that most of these currently used pesticides are generally not detected. However, there were also sporadic detections in surface water of thiophanate-methyl and carbendazim in lycium chinensis production areas, dicamba in rapeseed and garlic production areas, and acetamiprid in garlic and lycium chinensis production areas, and frequent detections of chlorpyrifos in surface water of rapeseed production areas in Huzhu County (detection rate of 71%), with a detected concentration range of 0.13~1.17 ng/L.

In recent years, most of the research on pesticide residues in water bodies in China has focused on the detection and assessment of organochlorine pesticides; evaluations of common pesticides in current use were relatively rare. Ji et al. [27] detected 39 target pesticides in water samples of Taihu Lake in Suzhou using a gas chromatography-mass spectrometry and found that only two herbicides, prometryn and terbutylhylazine, were detected, and the highest detected concentration was 0.75 μg/L. Yang et al. [20] used solid-phase extraction coupled with LC-MS/MS to detect 18 currently used pesticides, including thiamethoxam, imidacloprid and chlorpyrifos, in water from rivers and reservoirs around Tai’an City, and found no pesticide residues (all below the detection limits of 10 ng/L). Herbicides such as atrazine, 2,4-D-butyl, metribuzin, oxadione and oxyfluorfen were detected in water samples from the Harbin section of the Songhua River at certain concentrations, with the highest detection rate being atrazine, which was detected in 100% of the water bodies at levels ranging from 11.2 to 1671 ng/L [28].These results indicate that the current level of contamination of pesticides in water bodies is much lower than the national drinking water hygiene standard of 5.0 μg/L (GB 5749-2022). Meanwhile, these pesticides are non-persistent and have slight bioenrichments, which may result in no serious environmental risks. 

#### 3.1.2. Pollution Characteristics and Source Apportionment of the OCPs

As can be seen from Figure 2A,B, the two typical OCPs, i.e., HCHs and DDTs, were commonly detected in surface water from the four crop production areas. The residual concentrations of HCHs and DDTs in the water bodies had ranges of 0~1.68 ng/L and 0.41~2.41 ng/L, respectively. The mean values of HCHs in surface water of rapeseed, garlic, lycium chinensis and hulless barley production areas were 0.98, 1.02, 0.78 and 0.77 ng/L, respectively, while the mean values of residues of DDTs were 1.26, 1.41, 1.38 and 1.33 ng/L. The mean values of residues of DDTs in the water of the four production areas were higher than those of HCHs, yet the difference is not significant. As shown in Table 2, the pesticide residues of both OCPs in surface water were well below the environmental quality standard limits for surface water (GB3838-2002). Previous studies have shown that the OCPs in the water of lakes such as Chaohu, Poyang, Honghu and Baiyangdian in China are mainly HCHs and DDTs, with residues below 59.7 ng/L and 33.4 ng/L, respectively [6,8,29]. Earlier foreign studies found that total HCHs and DDTs in lake water could reach 400 ng/L and 560 ng/L, respectively [7,30,31], much higher than recent levels of OCPs in domestic water bodies. Although this is closely related to the local dosage, climate and water conditions, it also roughly reflects the obvious attenuation of OCPs with the passage of time. Noteworthily, some scholars have detected concentrations of DDTs in freshly fallen snow water from glaciers on Mount Everest at 0.42–1.64 ng/L of DDTs [32], which is comparable to the residue levels in the water bodies of recent studies and in the farmland water bodies of Qinghai in this study. This suggests that OCPs can be transported to the Qinghai–Tibet Plateau via the atmosphere, except for farmland applications.

The residual components of HCHs and DDTs in surface water from different crop production were considerably different. As shown in Figure 2C, *β*-HCH and *γ*-HCH were detected in significant amounts in surface waters of all four crop production areas, with *β*-HCH predominating. *α*-HCH was not detected in surface water from lycium chinensis production areas, while *δ*-HCH was not detected in surface water from garlic and rapeseed production areas. *β*-HCH is the most difficult of the HCHs isomers to degrade due to its own special structure that all chlorine atoms are in the carbon frame plane [29]. Meanwhile, the ratio of *β*-HCH/(*α*-HCH + *γ*-HCH) in all water bodies in this study is much greater than 0.5, indicating that there has been no new input of HCHs and lindane (*γ*-HCH) in surface waters of these crop production areas in the recent past [33]. For DDTs (Figure 2D), *pp’*-DDE contents were more than 50% in all four crop production areas waters, followed by *pp’*-DDT contents in the range of 10–30% in each crop production areas water. Meanwhile, *op’*-DDT was detected in all the different water bodies. In contrast, the proportion of *pp’*-DDD and *op’*-DDD was relatively low. DDTs residues in farmland mainly come from the use of DDTs and dicofol. The proportion of *pp’*-DDT and *op’*-DDT in the fraction of pesticide DDTs was 80–85% and 15–20% respectively, while the proportion of both components in dicofol was 1.7% and 11.4% respectively [34]. In the natural environment, DDTs are degraded to DDE and DDD under aerobic and anaerobic conditions, respectively [35]. The value of (DDD + DDE)/∑DDTs is commonly used in environmental pollution assessment to characterize the degradation pathways of DDTs and to analyze the sources of pollution [36]. The ratio of (DDD + DDE)/∑DDTs in this study was much higher than 0.5, indicating that the residues of DDTs in surface water of the four crop production areas in Qinghai mainly originated from historical doses that had degraded over a long period of time.

### 3.2. Analysis of Surface Water Nitrogen and Phosphorus Pollution Characteristics and Sources in Typical Crop-Producing Areas in Qinghai Province

The main physical and chemical indicators of surface water quality in four typical crop-producing areas in Qinghai Province are listed in Table 1. As can be seen, the four regional waters are weakly alkaline, with pH values ranging from 7.44 to 9.88, and no significant variability among regions. COD pollution shows a sporadic distribution, occurring in the irrigation canals of Dayinpo and Maji villages in the rapeseed production areas and Demang and Heini villages in the hulless barley production area, with concentrations ranging from 32.96 to 148.55 mg/L, comparable to the level of COD pollution in the surface water of the irrigation area of the Yellow River diversion in Ningxia [37]. This may be due to the fact that the water canals in the four villages pass through the villages and are polluted by the domestic sewage and farming effluents [37,38].

From the test results (Table 1), it can be seen that the mean values of TN, NO_3_^−^-N and NH_4_^+^-N concentrations were 2.95, 1.71 and 0.17 mg/L, respectively, across all the surface water from the four typical crop areas in Qinghai Province. Compared with the national surface water environmental quality standard (GB3838-2002) (Table 2), the TN concentration in 57.7% of the sampling sites exceeded the Class V water standard, among which those of all the samples in the garlic production area exceeded the Class V water standard. Only 15.4% of the water samples have TN concentrations above the Class III water standard, which are mainly located in hulless barley production areas, which have a 50% rate of Class III water compliance. Except for the point 25-DM in the hulless barley production area where NH_4_^+^-N concentrations were more than 22 times higher than NO_3_^−^-N concentrations, all other points were dominated by NO_3_^−^-N, with mean concentrations more than 70 times higher than NH_4_^+^-N. However, overall there were no significant differences in TN, NO_3_^−^-N and NH_4_^+^-N concentrations in the waters of the four crop-producing areas. The correlation analysis of pH, nitrogen and phosphorus concentrations in the water bodies (Figure 3A,B) showed that the NO_3_^−^-N and NH_4_^+^-N concentrations in the water bodies of the four crop production areas showed significant positive correlation with TN concentration (*p* < 0.05), which indicated that the nitrogen sources in the surface water of the four crop areas were relatively consistent. Meanwhile, the water bodies were eutrophic, and the nitrifying bacteria in the water bodies could convert part of the ammonium nitrogen into nitrate nitrogen [39]. Nitrogen pollution in water bodies in the four crop-producing areas of Qinghai is serious, with levels of all forms of nitrogen higher than those in the Longjiang River [11], the Xiangxi River [40] and Longtan Lake [41], and comparable to levels in the Ink River [42], the Guangzhou section of the Pearl River [43] and the Jiulong River [44]. NO_3_^−^-N concentrations were much higher than NH_4_^+^-N concentrations mainly because most of the water bodies investigated were flowing irrigation canals and rivers, which were rich in oxygen and favored the occurrence of nitrification [45]. In addition, the factors contributing to the variation in water nitrogen concentrations also include the point source pollution discharges and the activities of aquatic organisms around the environment [11]. The mean concentration of TP in water bodies in the four typical crop areas in Qinghai Province was 0.034 mg/L, with concentrations ranging from 0.01 to 0.07 mg/L. There were no significant differences in water TP concentrations among the production areas. According to the national environmental quality standard for surface water (GB3838-2002) (Table 2), these water bodies are classified as Class II water bodies in terms of TP concentrations. However, the distribution of water TP concentrations within each production area varied considerably. The TP concentrations at all points in the garlic production area were above 0.04 mg/L. However, only the water TP concentration at point 30-HN in the hulless barley production area was 0.06 mg/L, while at all other points they were below 0.03 mg/L. In the rapeseed production area, only point 9-DS and 1-DYP also had water TP concentrations of 0.04 and 0.06 mg/L, respectively, while in all other sampling sites they were below 0.03 mg/L. The TP concentrations in the surface water of lycium chinensis production areas were more discrete. The TP concentrations in Poyang Lake during the dry period ranged from 0.07 to 0.34 mg/L, with a mean concentration of 0.22 mg/L [46], which was significantly higher than the water bodies TP concentration in this study. The TP concentrations in the Longjiang River were 0.043 and 0.072 mg/L during the rich and dry periods, respectively [11], which were comparable to the water bodies in this research.

N/P values are important for evaluating the occurrence of ecological risks in water bodies. N/P values lower than seven in fresh water are nitrogen-limited, and greater than seven are phosphorus-limited [47,48]. At the same time, studies have shown that when TN < 0.2 mg/L and TP < 0.02 mg/L in the lake water, it is in a poor nutrient state; while, when TN > 0.2 mg/L and TP > 0.02 mg/L in the lake water, the water body reaches a eutrophic state [49]. In the four crop-producing areas of Qinghai Province of our study, as the N/P values ranged from 9.2 to 302 in the surface waters, phosphorus is a limiting factor for the proliferation of plankton in the water bodies. Therefore, reducing the entry of phosphorus into these areas may be the key to preventing continued deterioration of water quality. The excessive TN is mainly caused by nitrate nitrogen, which is mainly caused by the large amount of fertilizer application and the high degree of grazing activities in Qinghai Province. Agriculture and animal husbandry are relatively developed in Qinghai Province. Livestock activities around agricultural areas are frequent and overgrazing is common. A large amount of livestock and poultry excrement is generated into the watershed along with streams and rainwater, causing pollution to the water bodies in the basin. Meanwhile, the rapid development of tourism in Qinghai Province in recent years, human activities and vehicle emissions are also important reasons for the TN exceedance in the region.

### 3.3. Correlation Analysis of OCPs, and Nitrogen and Phosphorus Pollution in Surface Waters of Typical Crop-Producing Areas in Qinghai Province

OCPs were commonly detected in water bodies from areas producing rapeseed, garlic, lycium chinensis and hulless barley in Qinghai Province, and HCHs and DDTs were mostly present as *β*-HCH and *pp’*-DDE in water bodies. According to the national environmental quality standard for surface water (GB3838-2002), TN concentrations in these sampling sites were most above the standard of Class V, indicating that most surface water was in eutrophic condition. Correlation analysis of OCPs with physicochemical indicators pH, NH_4_^+^-N, NO_3_^−^-N, TN and TP revealed that HCHs showed significant negative and positive correlations with NO_3_^−^-N and TP, respectively (Figure 3C,D). The degradation of OCPs in water bodies mainly occurs through the biodegradation via microorganisms and algae. The pollution of nitrogen in the water bodies of the study area is mainly in the form of NO_3_^-^-N, and the presence of large amounts of nitrogen leads to the eutrophication of the water bodies, which also resulted in the proliferation of water microorganisms and algae, thus accelerating the degradation of HCHs. In this research, the water bodies’ N/P ratio at most sites was greater than seven, showing a phosphorus-limited state, which is a nutrient condition favorable to the survival of algae [50]. In contrast, there was no significant correlation between DDTs and indicators such as nitrogen and phosphorus, which may explain the more complex source of DDTs and the longer persistence in water [29].

## 4. Conclusions

(1) The results show that most of these currently used pesticides were generally not detected. However, there were also sporadic detections in surface water of thiophanate-methyl and carbendazim in lycium chinensis production areas, dicamba in rapeseed and garlic production areas, and acetamiprid in garlic and lycium chinensis production areas, and frequent detections of chlorpyrifos in the surface water of rapeseed production areas, with a detected concentration range of 0.13~1.17 ng/L. These results are far below the national drinking water hygiene standard of 5.0 μg/L (GB 5749-2022). Since these pesticides are non-persistent pollutants, their bioenrichment capacity is significantly lower than that of OCPs. Therefore, their environmental risk is much lower than that of OCPs.

(2) HCHs and DDTs were commonly detected in water bodies from the four specialty crop production areas in Qinghai Province, with no significant differences among the four crop-producing areas. The detected concentrations ranged from 0 to 1.68 ng/L and 0.41 to 2.41 ng/L, respectively, with low levels of pollution. Source analysis of HCHs and DDTs revealed that HCHs and DDTs were mainly present as *β*-HCH and *pp’*-DDE, and that residual contamination of both OCPs was due to historical application residues. However, given their apparent bioconcentration, the OCPs with low-level residues still pose a potential ecological risk to water bodies in these regions.

(3) All four crop-producing areas showed weak alkalinity in the water bodies, with COD pollution detected at individual points. There were no significant differences in the concentrations of TN, NO_3_^−^-N, NH_4_^+^-N and TP between the four areas, with mean concentrations of 2.95, 1.71, 0.17 and 0.034 mg/L, respectively, with NO_3_^−^-N being the main form of nitrogen present. Referring to the national surface water environmental quality standard (GB3838-2002), the TN concentration in 57.7% of the points exceeds the water standard of Class V, and the nitrogen pollution in the water bodies of garlic-producing areas is relatively serious. With N/P values ranging from 9.2 to 302 in the four crop-producing regions of Qinghai Province, the water bodies are in a phosphorus-limited state, so reducing the discharge of elemental phosphorus in these areas is crucial to preventing continued deterioration of water quality.

(4) HCHs showed a significant negative correlation with NO_3_^−^-N content and a significant positive correlation with TP concentration. The degradation of OCPs in water bodies is mainly through the degradation of water body microorganisms and algae. The pollution of nitrogen in the water bodies of the study area occurs mainly in the form of NO_3_^−^-N, and the large amounts of nitrogen leads to the eutrophication of the water bodies, which also allows the proliferation of microorganisms and algae in water, thus accelerating the degradation of HCHs.

## Figures and Tables

**Figure 1 ijerph-19-16392-f001:**
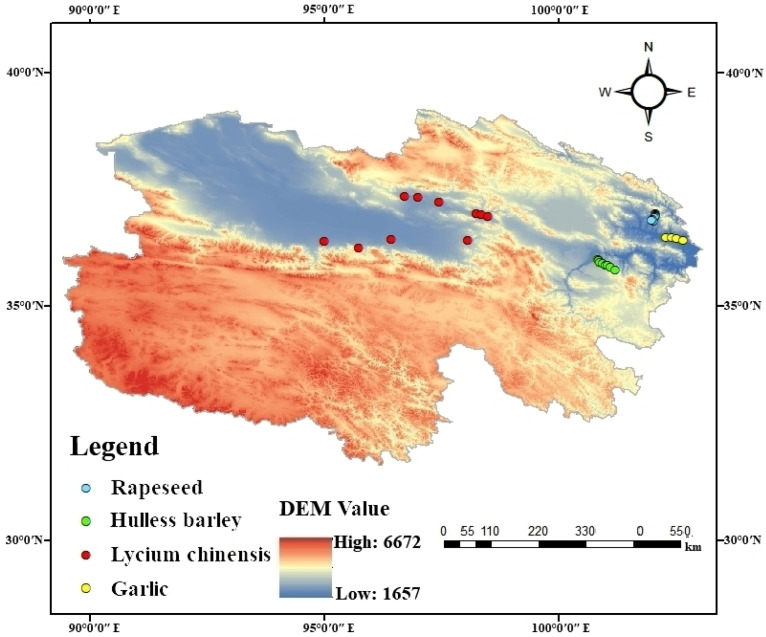
Schematic diagram of the sampling points for surface water in four typical crop-producing areas.

**Figure 2 ijerph-19-16392-f002:**
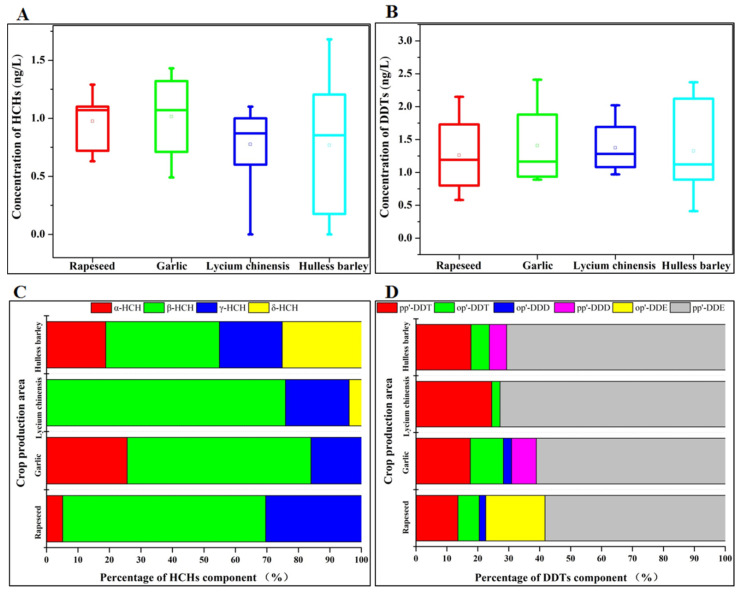
Residue status of organochlorine pesticides (HCHs and DDTs) in surface water samples from four typical crop areas. (**A**) Residue Levels of HCHs; (**B**) Residue Levels of DDTs; (**C**) Residue composition of HCHs; (**D**) Residue composition of DDTs.

**Figure 3 ijerph-19-16392-f003:**
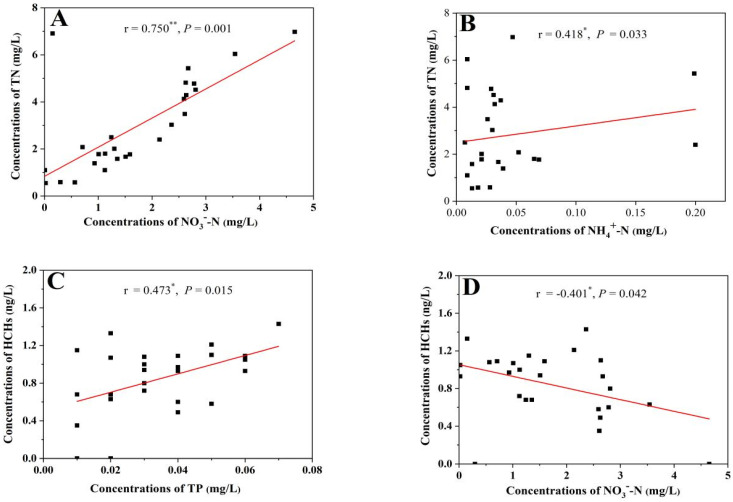
Correlation analysis of physicochemical indices and residues of HCHs in surface water samples from four typical crop-producing areas: (**A**) extremely positive correlation between the total nitrogen concentration and the nitrate concentration; (**B**) positive correlation between the total nitrogen concentration and the ammonium concentration; (**C**) negative correlation between the residues of HCHs and the nitrate concentration; (**D**) positive correlation between the residues of HCHs and the total P.

**Table 1 ijerph-19-16392-t001:** Physicochemical indexes of water quality in typical crop-producing areas of Qinghai Province.

Crop Production Areas	Latitude and Longitude	Water Number	pH	COD mg/L	TN mg/L	TP mg/L	NO_3_^−^ mg/L	NH_4_^+^ mg/L
Rapeseed in Huzhu County, Haidong City	102.076, 36.976	1-DYP	7.44	148.55	0.55	0.06	0.020	0.013
	102.073, 36.955	2-DHL	8.30	0	1.10	0.03	1.123	0.009
	102.066, 36.940	3-SC	8.32	0	1.78	0.02	1.008	0.021
	102.049, 36.910	4-YJZ	8.08	0	2.01	0.01	1.299	0.021
	102.044, 36.894	6-MJ	9.88	38.41	6.04	0.02	3.543	0.009
	102.003, 36.861	8-LRT	8.15	0	2.50	0.02	1.243	0.007
	101.980, 36.838	9-DS	9.43	0	2.08	0.04	0.708	0.052
Garlic in Ledu District, Haidong City	102.299, 36.480	10-HT	9.88	0	4.82	0.04	2.626	0.009
	102.427, 36.474	11-TQ	8.11	0	5.43	0.04	2.672	0.199
	102.528, 36.447	12-SY	7.96	0	3.03	0.07	2.362	0.030
	102.662, 36.405	13-LJ	8.43	0	2.40	0.05	2.139	0.200
Lycium chinensis in Nuomuhong Prefecture, Haixi Prefecture	96.433, 36.437	14-CB	8.49	0	1.67	0.03	1.507	0.035
	95.005, 36.395	15-GLM	8.64	0	1.77	0.06	1.589	0.069
	95.730, 36.243	16-DGL	8.86	0	1.58	0.01	1.354	0.013
	98.062, 36.402	17-SZY	8.22	0	4.13	0.05	2.594	0.032
	96.716, 37.344	18-HTT	8.20	0	4.29	0.05	2.636	0.037
	97.453, 37.221	20-QS	7.98	0	4.78	0.04	2.782	0.029
	98.241, 36.988	21-XC	8.23	0	4.52	0.03	2.810	0.031
	98.350, 36.957	22-TH	8.41	0	1.39	0.04	0.931	0.039
	98.502, 36.918	23-DZ	7.87	0	1.80	0.03	1.127	0.065
Hulless barley in Guinan County, Hainan Prefecture	100.834, 36.003	24-GT	8.10	0	6.98	0.01	4.653	0.047
	100.857, 35.977	25-DM	8.13	32.96	6.91	0.02	0.150	3.325
	100.879, 35.945	27-GL	8.25	0	0.59	0.02	0.296	0.028
	100.938 35.912	28-LZ	8.33	0	3.49	0.01	2.609	0.026
	101.000, 35.889	29-LHX	8.26	0	0.58	0.03	0.565	0.018
	101.057, 35.877	30-HN	7.44	148.55	0.55	0.06	0.020	0.013

**Table 2 ijerph-19-16392-t002:** Surface water quality monitoring standards (GB3838-2002).

Indicator	Class I	Class II	Class III	Class IV	Class V	Analysis Methods	Minimum Detection Limit
COD (mg/L)	≤15	≤15	≤20	≤30	≤40	Dichromate method (GB11914-89)	5 mg/L
TN(mg/L)	≤0.2	≤0.5	≤1.0	≤1.5	≤2.0	alkaline potassium persulphate digestion UV spectrophotometric method (GB11914-89)	0.05 mg/L
NH_4_^+^-N(mg/L)	≤0.15	≤0.5	≤1.0	≤1.5	≤2.0	Salicylic acid spectrophotometry (GB 7481-87)	0.01 mg/L
TP(mg/L)	≤0.02	≤0.1	≤0.2	≤0.3	≤0.4	Ammonium molybdate spectrophotometer (GB11893-89)	0.01 mg/L
NO_3_^−^-N (mg/L)	≤10 (Standard limits)					Ion chromatography (HJ/T 84-2001)	0.08 mg/L
pH (dimensionless)	6~9 (Standard limits)					Glass electrode method (GB6920-86)	
Lindane(γ-HCH) (μg/L)	≤2.0 (Standard limits)					Gas chromatography (GB 7492-87)	0.1 ng/L
DDTs (μg/L)	≤1.0 (Standard limits)					Gas chromatography (GB 7492-87)	0.1 ng/L

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
