# Peer review of "Analysis of Pollution Characteristics and Sources in Surface Water in Typical Crop-Producing Areas of Qinghai Province"

_ijerph, 2022, doi:10.3390/ijerph192416392_

Round 1

Reviewer 1 Report

The article is very interesting.

The measuring  concentration all kind of pesticides (old eq. DDT or modern eq. carbendazim) are very important.

The results are also interesting specially for DDT or similar pesticides – it is find all time, not at big amount but found.

Once again it pointed that modern pesticides are safe for people health.

All conclusions are good.

In my opinion the testing COD  are not did in good resolution aparature:

Table 2, 10 mg/L limit of detection much better would be 1 mg/L.

In fact if COD < 50 mg/L the standard of surface ware are quite good.

The authors should have  think once again about this problem, explain it more carefully the problem with COD.

In the Table 1 the average concentration for different region not quite good expresses the state of water in the region, the region is too big, it should be done for much smaller area or rejected from paper.

The next problem is for TN and TP, it  pointed that in the future would be appear problem with the biological structure of life in surface water it would be  changed it is a new  problem for very intensive agriculture .  

If we will consider  that hulless barley  and  rapeseed  should have good type of soil and demand intensive fertilization for high yields we may have cause of bigger COD or TN in surface water.

Author Response

We sincerely appreciate the encouragement and suggestions of the reviewer.  The point to point responses are as follows,

In my opinion, the testing COD are not did in good resolution aparature: Table 2, 10 mg/L limit of detection much better would be 1 mg/L. In fact if COD < 50 mg/L the standard of surface ware are quite good. The authors should have thought once again about this problem, explain it more carefully the problem with COD.

Response: Thank you for your good suggestions. However, the limit of detection for COD is 5 mg/L according to HJ 828-2017. The change has been noted in the manuscript.

In the Table 1, the average concentration for different region not quite good expresses the state of water in the region, the region is too big, it should be done for much smaller area or rejected from paper.

Response: The average concentrations have been removed as you requested, and the latitude and longitude have been used instead of the village name for greater rigor. See the revised manuscript for details, please.

The next problem is for TN and TP, it  pointed that in the future would be appear problem with the biological structure of life in surface water it would be  changed it is a new  problem for very intensive agriculture.  If we will consider that hulless barley and rapeseed should have good type of soil and demand intensive fertilization for high yields we may have cause of bigger COD or TN in surface water.

Response: Intensive fertilization will increase crop yields while increasing COD or TN content. Therefore, scientific fertilization technology can be used to improve fertilizer varieties, improve fertilization methods and reasonable irrigation technology to make full use of the purification capacity of land and vegetation. Strengthen the circulation of nitrogen and other substances in the terrestrial ecosystem, and finally control and reduce the contents of total nitrogen, total phosphorus and COD in the water, and reduce the eutrophication of water.

Reviewer 2 Report

The review concerned the scientific article: Analysis of Pollution Characteristics and Sources in Surface Water in Typical Crop-Producing Areas of Qinghai Province.

An important topic from the perspective of broadly understood environmental protection.

Introduction written correctly. In the material and methods chapter, the authors specify the number of samples taken from which places. Please explain what guided the selection of the number of samples taken from a given place? The description of the analyzes carried out and their explanation, in my opinion, is exhaustively written.

The results and discussion chapters have been merged. This makes the interpretation of the presented results a bit difficult, but it is not a mistake. The test results are described in great detail and accurately. Despite the combination of chapters, the authors used divisions into sections, which makes it much easier to read. The literature cited in the discussion comes from recent years and is up to date. Appropriate conclusions.

I accept for printing.

Author Response

We sincerely appreciate the encouragement and suggestions of the reviewer.  The point to point responses are as follows.

An important topic from the perspective of broadly understood environmental protection.

Introduction is written correctly. In the material and methods chapter, the authors specify the number of samples taken from which places. Please explain what guided the selection of the number of samples taken from a given place? The description of the analyzes carried out and their explanation, in my opinion, is exhaustively written.

Response: Importantly, the monitoring points should be set, and the plots with strong representativeness and potential pollution should be selected in accordance with the principle that they can represent the whole monitoring area of origin. For the field planting area, if the producing area is less than 500 hm2, there are 3 layout points; if the producing area is 500 hm2 ~ 2000 hm2, there are 5 layout points; if the producing area is more than 2000 hm2, one sampling point will be added for every 1000 hm2 increase. Different origin types of water will also have different sampling points. According to the size of the growing area and the type of water in the producing area, different numbers of monitoring points are set for sampling.

The results and discussion chapters have been merged. This makes the interpretation of the presented results a bit difficult, but it is not a mistake. The test results are described in great detail and accurately. Despite the combination of chapters, the authors used divisions into sections, which makes it much easier to read. The literature cited in the discussion comes from recent years and is up to date. Appropriate conclusions.

Response: As you said, the combination of results and discussion in this article can make the full text more compact and clear, and this format is also allowed by the journal.